# Transitioning the Molecular Tumor Board from Proof of Concept to Clinical Routine: A German Single-Center Analysis

**DOI:** 10.3390/cancers13051151

**Published:** 2021-03-08

**Authors:** Rouven Hoefflin, Adriana Lazarou, Maria Elena Hess, Meike Reiser, Julius Wehrle, Patrick Metzger, Anna Verena Frey, Heiko Becker, Konrad Aumann, Kai Berner, Martin Boeker, Nico Buettner, Christine Dierks, Jesus Duque-Afonso, Michel Eisenblaetter, Thalia Erbes, Ralph Fritsch, Isabell Xiang Ge, Anna-Lena Geißler, Markus Grabbert, Steffen Heeg, Dieter Henrik Heiland, Simone Hettmer, Gian Kayser, Alexander Keller, Anita Kleiber, Alexandra Kutilina, Leman Mehmed, Frank Meiss, Philipp Poxleitner, Justyna Rawluk, Juri Ruf, Henning Schäfer, Florian Scherer, Khalid Shoumariyeh, Andreas Tzschach, Christoph Peters, Tilman Brummer, Martin Werner, Justus Duyster, Silke Lassmann, Cornelius Miething, Melanie Boerries, Anna L. Illert, Nikolas von Bubnoff

**Affiliations:** 1Department of Medicine I, Medical Center, University of Freiburg, Faculty of Medicine, University of Freiburg, 79106 Freiburg, Germany; rouven.hoefflin@uniklinik-freiburg.de (R.H.); adriana.lazarou@uniklinik-freiburg.de (A.L.); julius.wehrle@uniklinik-freiburg.de (J.W.); heiko.becker@uniklinik-freiburg.de (H.B.); christine.dierks@uniklinik-freiburg.de (C.D.); jesus.duque.afonso@uniklinik-freiburg.de (J.D.-A.); ralph.fritsch@uniklinik-freiburg.de (R.F.); alexander.keller@uniklinik-freiburg.de (A.K.); anita.kleiber@uniklinik-freiburg.de (A.K.); alexandra.kutilina@uniklinik-freiburg.de (A.K.); justyna.rawluk@uniklinik-freiburg.de (J.R.); florian.scherer@uniklinik-freiburg.de (F.S.); khalid.shoumariyeh@uniklinik-freiburg.de (K.S.); justus.duyster@uniklinik-freiburg.de (J.D.); cornelius.miething@uniklinik-freiburg.de (C.M.); Nikolas.Bubnoff@uniklinik-freiburg.de (N.v.B.); 2Comprehensive Cancer Center Freiburg, Medical Center, University of Freiburg, Faculty of Medicine, University of Freiburg, 79106 Freiburg, Germany; maria.elena.hess@uniklinik-freiburg.de (M.E.H.); meike.reiser@uniklinik-freiburg.de (M.R.); anna.frey@uniklinik-freiburg.de (A.V.F.); konrad.aumann@uniklinik-freiburg.de (K.A.); kai.berner@uniklinik-freiburg.de (K.B.); martin.boeker@imbi.uni-freiburg.de (M.B.); nico.buettner@uniklinik-freiburg.de (N.B.); michel.eisenblaetter@uniklinik-freiburg.de (M.E.); thalia.erbes@uniklinik-freiburg.de (T.E.); isabell.xiang.ge@uniklinik-freiburg.de (I.X.G.); annalena.geissler1@gmail.com (A.-L.G.); markus.grabbert@uniklinik-freiburg.de (M.G.); steffen.heeg@uniklinik-freiburg.de (S.H.); dieter.henrik.heiland@uniklinik-freiburg.de (D.H.H.); simone.hettmer@uniklinik-freiburg.de (S.H.); gian.kayser@uniklinik-freiburg.de (G.K.); leman.mehmed@uniklinik-freiburg.de (L.M.); frank.meiss@uniklinik-freiburg.de (F.M.); philipp.poxleitner@uniklinik-freiburg.de (P.P.); juri.ruf@uniklinik-freiburg.de (J.R.); henning.schaefer@uniklinik-freiburg.de (H.S.); andreas.tzschach@uniklinik-freiburg.de (A.T.); christoph.peters@mol-med.uni-freiburg.de (C.P.); tilman.brummer@mol-med.uni-freiburg.de (T.B.); martin.werner@uniklinik-freiburg.de (M.W.); silke.lassmann@uniklinik-freiburg.de (S.L.); melanie.boerries@uniklinik-freiburg.de (M.B.); 3Institute of Medical Bioinformatics and Systems Medicine, Medical Center, University of Freiburg, Faculty of Medicine, University of Freiburg, 79106 Freiburg, Germany; patrick.metzger@uniklinik-freiburg.de; 4Institute for Surgical Pathology, Medical Center, University of Freiburg, Faculty of Medicine, University of Freiburg, 79106 Freiburg, Germany; 5German Cancer Consortium (DKTK), Partner Site Freiburg and German Cancer Research Center (DKFZ), 69120 Heidelberg, Germany; 6Department of Obstetrics and Gynaecology, Medical Center, University of Freiburg, Faculty of Medicine, University of Freiburg, 79106 Freiburg, Germany; 7Institute for Medical Biometry and Statistics, Medical Center, University of Freiburg, Faculty of Medicine, University of Freiburg, 79106 Freiburg, Germany; 8Department of Medicine II, Medical Center, University of Freiburg, Faculty of Medicine, University of Freiburg, 79106 Freiburg, Germany; 9Freiburg, Department of Radiology, Medical Center, University of Freiburg, Faculty of Medicine, University of Freiburg, 79106 Freiburg, Germany; 10Department of Urology, Medical Center, University of Freiburg, Faculty of Medicine, University of Freiburg, 79106 Freiburg, Germany; 11Department of Neurosurgery, Medical Center, University of Freiburg, Faculty of Medicine, University of Freiburg, 79106 Freiburg, Germany; 12Department of Pediatric Hematology and Oncology, Medical Center, University of Freiburg, Faculty of Medicine, University of Freiburg, 79106 Freiburg, Germany; 13Clinical Cancer Registry, Medical Center, University of Freiburg, Faculty of Medicine, University of Freiburg, 79106 Freiburg, Germany; 14Department of Dermatology and Venerology, Medical Center, University of Freiburg, Faculty of Medicine, University of Freiburg, 79106 Freiburg, Germany; 15Department of Oral and Maxillofacial Surgery, Medical Center, University of Freiburg, Faculty of Medicine, University of Freiburg, 79106 Freiburg, Germany; 16Department of Nuclear Medicine, Medical Center, University of Freiburg, Faculty of Medicine, University of Freiburg, 79106 Freiburg, Germany; 17Department of Radiation Oncology, Medical Center, University of Freiburg, Faculty of Medicine, University of Freiburg, 79106 Freiburg, Germany; 18Institute of Human Genetics, Medical Center, University of Freiburg, Faculty of Medicine, University of Freiburg, 79106 Freiburg, Germany; 19Institute of Molecular Medicine and Cell Research, Faculty of Medicine, University of Freiburg, 79104 Freiburg, Germany; 20Center for Biological Signaling Studies BIOSS, University of Freiburg, 79104 Freiburg, Germany

**Keywords:** personalized cancer medicine, precision oncology, molecular profiling, molecular tumor board, targeted therapies, combination therapies, cancer progression, cancer genetics, cancer molecular biology, cancer immunotherapy

## Abstract

**Simple Summary:**

Access to molecular cancer treatments outside of clinical trials is limited and the benefit of molecular-guided, individualized patient care in patients with cancer progression after standard treatment is unclear. We here present the four-year experience of one of Europe’s first Molecular Tumor Boards and show that precision oncology in the era of affordable, extended genetic and phenotypic tumor profiling is feasible and effective for a small but relevant proportion of advanced cancer patients. We performed a comprehensive analysis of clinical follow-up data and report our workflow optimizations and upscaling processes. These could help other centers to establish similar structures to support molecular-guided treatment for patients with limited therapy options.

**Abstract:**

Molecular precision oncology faces two major challenges: first, to identify relevant and actionable molecular variants in a rapidly changing field and second, to provide access to a broad patient population. Here, we report a four-year experience of the Molecular Tumor Board (MTB) of the Comprehensive Cancer Center Freiburg (Germany) including workflows and process optimizations. This retrospective single-center study includes data on 488 patients enrolled in the MTB from February 2015 through December 2018. Recommendations include individual molecular diagnostics, molecular stratified therapies, assessment of treatment adherence and patient outcomes including overall survival. The majority of MTB patients presented with stage IV oncologic malignancies (90.6%) and underwent an average of 2.1 previous lines of therapy. Individual diagnostic recommendations were given to 487 patients (99.8%). A treatment recommendation was given in 264 of all cases (54.1%) which included a molecularly matched treatment in 212 patients (43.4%). The 264 treatment recommendations were implemented in 76 patients (28.8%). Stable disease was observed in 19 patients (25.0%), 17 had partial response (22.4%) and five showed a complete remission (6.6%). An objective response was achieved in 28.9% of cases with implemented recommendations and for 4.5% of the total population (22 of 488 patients). By optimizing the MTB workflow, case-discussions per session increased significantly while treatment adherence and outcome remained stable over time. Our data demonstrate the feasibility and effectiveness of molecular-guided personalized therapy for cancer patients in a clinical routine setting showing a low but robust and durable disease control rate over time.

## 1. Introduction

Oncology evolves as the most innovative field in medicine; it was responsible for 28 FDA drug approvals from 2018 to 2019, a rapid expansion of the anti-cancer drug repertoire comprising numerous targeted and immunotherapies [1,2]. For the first time, approvals have been granted to tumor-agnostic drugs including pembrolizumab for microsatellite instability-high (MSI-H), mismatch repair deficient (MMRd), or tumor mutational burden-high (TMB-H) metastatic tumors [3,4,5] and larotrectinib for malignancies harboring neurotrophic receptor tyrosine kinase (*NTRK*) gene fusions [6]. These approvals highlight the increasing therapeutic significance of molecular and genetic testing in oncology. After initial trials failed to demonstrate a clear benefit [7,8,9,10], numerous studies and reports now support the feasibility and the effectiveness of molecular precision oncology [11,12,13,14,15,16,17]. Despite these advances in tailored diagnostic and therapy, cancer remains the second leading cause of death globally, accounting for over 9 million deaths world-wide in 2018 [18].

The large number of available cancer treatments and clinical trials in combination with increasing accessibility of affordable extended molecular tumor profiling offer numerous new therapy options, especially for patients who have failed standard-of-care treatment. Furthermore, the growing knowledge of predictive biomarkers and the understanding of treatment resistance mechanisms rapidly change treatment paradigms. To keep pace with the dynamic field of personalized precision oncology and to provide cancer patients state-of-the-art molecular diagnostics and treatment recommendations, we established a Molecular Tumor Board (MTB) at the University Medical Center Freiburg Comprehensive Cancer Center in March 2015. Here, we report a broad four-year analysis showing patient characteristics, diagnostics and therapy recommendations, adherence, and clinical outcome measurements. Following and extending the report of our proof-of-concept study in 2018 [19], our most recent analyses focus on structural and organizational improvements that enabled us to triple the annual cases while confirming that clinical actionability of molecular targets translates into improved patient outcomes.

## 2. Materials and Methods

### 2.1. MTB Organization and Patients

The MTB was founded in March 2015 and comprises a multidisciplinary team of physicians with molecular cancer expertise from more than 16 departments and experts from molecular pathology, molecular biology, and medical bioinformatics. There are no formal inclusion or exclusion criteria for the presentation of patients to the MTB. Thus, it is open for all cancer patients with the intention to focus on those who lack standard treatment options or suffer from rare tumors. All patients discussed from March 2015 to December 2018 and treated on site (*n* = 488) were included in this retrospective, single-center analysis. Patients are registered via an online system by the treating physician, who is responsible for the initial clinical case presentation (Appendix A). Recommendations for molecular analyses are given after the first case presentation according to entity-specific and entity-independent diagnostic standard operation procedures (SOP, Supplementary Procedures), that are regularly updated. Treatment recommendations are given upon interdisciplinary discussion according to results of molecular analyses, which are presented by the pathology- and bioinformatics-team at the second case presentation and include levels of molecular evidence (Appendix A). This trial was approved by the local institutional review board of the Medical Center—University of Freiburg (protocol code 369/19). All patients gave written informed consent.

### 2.2. Molecular Diagnostics

The MTB determines patient tissue sampling for molecular pathology analyses or recommends re-biopsies if necessary. All routine molecular analyses (RMA) are performed using nationally certified tests in the accredited laboratories of the Institute for Surgical Pathology. These include immunohistochemistry (IHC) (comprising an immuno-oncology (IO)-panel, mismatch repair deficiency (MMRd)-testing, NTRK-testing, and individual biomarkers), in situ hybridization, microsatellite instability (MSI)-testing, and targeted next generation sequencing (tNGS; Supplementary Procedures). The IO-panel includes IHC against CD3, CD4, CD8, PD1, and PD-L1. Scoring was performed by evaluating the combined positivity score (CPS), tumor proportion score (TPS), and the immune cell (IC) score [20]. MSI-testing is performed by microsatellite analysis using either the standard panel of two mononucleotide (BAT-25, BAT-26) and three dinucleotide (D2S123, D5S346, and D17S250) [21,22] or using a commercial test (Promega, Walldorf, Germany) of five monomorphic mononucleotide (BAT-25, BAT-26, MONO-27, NR-21, and NR-24) repeat markers and two polymorphic pentanucleotide (Penta C and Penta D) for additional quality control [23]. NTRK-testing is performed with IHC (NTRK-A, B, and C) and/or by *NTRK1*/*2*/*3*-RNA-Fusion hybrid-capture NGS analysis. Extended genetic analysis (EGA) includes whole-exome sequencing (WES) and RNA sequencing (RNA-Seq) and were performed using mostly formalin-fixed paraffin-embedded (FFPE) tissue (90 of 104 patients; 86.5%) or fresh-frozen samples (14 of 104 patients; 13.5%). WES is performed on DNA extracted from micro-dissected tumor tissue and complementary germline DNA from peripheral blood or healthy tissue to distinguish germline from somatic variants. The lowest tumor cell content was 10%. After quality control and trimming with Trimmomatic [24], variant calling is performed with VarScan2 [25,26] and subsequent false positive filtering. Only non-silent variants detected with a variant allele frequency (VAF) greater than 10% and reported with a population frequency less than 0.1% (MAF, minor allele frequency) in the Genome Aggregation Database (gnomAD) [27] are reported. Single nucleotide variations (SNVs) and small insertions and deletions (InDels) are classified according to ClinVar [28], InterVar [29], COSMIC [30], dbSNP [31], cancer hotspots [32,33], drug–gene interaction (DGIdb) [34], and functionally annotated according to the dbNSFP database [35,36] which contains 37 prediction and nine conservation scores. Mutation Signature Analysis is performed with the R/Bioconductor [37,38] package YAPSA [39] and the COSMIC signatures (v2). Copy number alteration analysis is performed using Control-FREEC [40,41]. TMB was calculated as the number of non-silent somatic mutations divided by the number of targeted regions in megabases and was only assessed for WES data. Tumor samples harboring more than 10 mutations per megabase were defined as TMB-H in accordance with the FDA approval of pembrolizumab for TMB-H solid tumors [5]. The BRCAness-score was calculated as the percentage of the AC3-signature in relation to all other mutational signatures classified by the YAPSA-package [39]. RNA fusions are identified with FusionCatcher [42] and the STAR aligner [43] is used to align and infer the gene expression level. The repertoire of tNGS-panels includes a hotspot eight-gene panel (Illumina, San Diego, CA, USA), a 15 gene panel (Appendix A), a hotspot 48-gene panel (TruSeq Amplicon Cancer Panel, Illumina) [44], a BRCA1/2 panel (Illumina), and a 54-gene myeloid panel (TruSight Myeloid Sequencing Panel, Illumina). Analyses are performed on tumor tissue only and are processed with the Illumina pipeline. Variants are additionally classified and reported to the entity experts according to MAF, cancer hotspots, COSMIC, dbSNP, and Condel [45] similar to variants detected in WES. Variants not comprised in the COSMIC- or cancer hotspots-database were categorized as “not annotated”. Their predicted clinical relevance was assessed by the variant characteristics mentioned above and literature research done by the entity experts.

## 3. Results

From March 2015 to December 2018, a total of 488 patients were discussed in the MTB with 1072 individual case presentations and an average of 2.2 discussions per patient. During 95 board meetings, an average of 16 clinicians from up to 16 departments and experts in molecular pathology, molecular biology, and medical bioinformatics participated in the sessions. By implementing entity-specific diagnostic SOP (Supplementary Procedures) in 2017, the number of cases per MTB-session (90 min) tripled from an average of 5.8 in 2015 to an average of 19.7 in 2018. The median turnaround time from initial case presentation to first treatment recommendation was 42 days.

### 3.1. Patient Characteristics

Median age at first MTB presentation was 54 years (range 1–88 years) with a similar distribution between females (47.1%) and males (52.9%). An overview of the patient characteristics is shown in Table 1. The vast majority of patients suffered from solid tumor malignancies (*n* = 470; 96.5%) in metastatic stage (*n* = 383; 78.5%). The most frequent tumor categories were lower gastrointestinal tract (*n* = 68; 13.9%), pancreas (*n* = 50; 10.2%), and central nervous system (*n* = 45; 9.2%). The median of previous lines of therapies was 2.1 (range 0–12) including a significant number of heavily pretreated patients that had received more than three prior therapies (*n* = 79; 16.2%). Although there were no formal inclusion and exclusion criteria, referral to the MTB was mostly initiated because of progression to standard of care treatment (*n* = 381, 78.1%) or rare tumor entities (*n* = 51; 10.5%).

### 3.2. Molecular Diagnostic Testing

Diagnostic recommendations were given to all but one patient (*n* = 487, 99.8%), who was diagnosed with Burkitt lymphoma and referred to the lymphoma board. Based on regularly updated diagnostic SOP we recommended both entity-specific as well as entity-independent tests. The latter includes MSI-, MMR-, IO-panel-, and NTRK-testing. Almost all patients received at least one recommendation for RMA (*n* = 485, 99.4%), whereas more than one third was recommended to undergo EGA (*n* = 183, 37.5%; Table 2). Overall, 615 of 762 diagnostic recommendations (80.7%) were pursued. Common causes for lack of implementation were technical reasons (i.e., lack of appropriate tumor tissue for RMA or insufficient isolation of tumor DNA for EGA; 44.9%), medical reasons (17.7%), or patient death (18.4%; Appendix A).

The majority of implemented RMA (*n* = 3550) were IHC (*n* = 2599), tNGS-panels (*n* = 412), and in situ hybridizations (*n* = 227; Appendix A). The most frequently used tNGS assay was a 48-gene panel (*n* = 221) that identified a total of 502 COSMIC annotated mutations (Figure 1A, Appendix A). The top five mutated genes were *APC*, *TP53*, *ATM*, *SMAD4,* and *ERBB4*. Variants with direct therapeutic implications (*n* = 343) were called using the OncoKB [46] database and were categorized as potentially drug-sensitizing mutations including *BRAF*, *IDH1*, *KIT,* and *PIK3CA* or drug-resistance variants in *KRAS*, *NRAS,* and *KIT*.

EGA was recommended to 183 patients (37.5%; WES and/or RNA-seq: *n* = 180; RNA-seq only: *n* = 3), especially to young individuals (<50 years), patients with rare tumors or in cases in which RMA did not identify a therapeutic target. Re-biopsies were recommended to 40 patients (8.2%) in cases in which adequate tumor material was not available (i.e., low amount of tissue, low tumor cell content, long time from biopsy to presentation). WES was performed for 104 patients (21.3%) and revealed 10,484 mutations, including 2064 COSMIC-annotated (thereof 64 in hotspot regions) and 8420 unknown somatic variants (Figure 1B,C, Appendix A). According to TARGET or DGIdb, 1987 mutations were classified as actionable. The more clinically relevant OncoKB classification revealed 53 targetable mutations (Figure 1C and Appendix A). An overview of the most commonly mutated genes, including the corresponding entities, is shown in Figure 1B. Cancer genes that were mutated in at least three cases (*n* = 19) are depicted in Appendix A. The top five mutated cancer genes were *TP53*, *KRAS*, *APC*, *BRAF,* and *AR*. The bioinformatics pipeline further reported copy number variations (CNV) from oncogenes (Figure 1D) and tumor suppressor genes (Appendix A). Of special interest were high copy number gains of oncogenes (>7; dark red in Figure 1D) that were annotated according to the OncoKB database and represent possible actionable targets (Gene names in red in Figure 1D). We also assessed TMB as a surrogate for neoantigen driven tumor immunogenicity and as a predictive marker of response to immune checkpoint blockade (ICB)-based therapy (Figure 1D). TMB-H (>10 mutations per megabase) was found in 10 of 104 sequenced patients (9.6%). Additionally, we determined mutational signatures that reflect different exogenous and/or endogenous mutational processes that cause somatic mutations in cancer (Appendix A) [47,48]. Of importance to the MTB, the AC3-signature reflects defects in homologous recombination repair mechanisms known from BRCA-deficient tumors [47,48,49]. Following the concept of synthetic lethality, tumors harboring this BRCAness phenotype might be susceptible towards poly-(ADP)-ribose polymerase inhibitors (PARPis) or DNA damaging agents [50,51,52]. A positive BRCAness-score (>20%, based on NCT03127215) was seen in 29 of 104 sequenced patients (27.9%; Figure 1B).

### 3.3. Treatment Recommendations

The interpretation and interdisciplinary discussion of diagnostic results led to a treatment recommendation in 264 of 488 patients (54.1%). A total of 367 treatment recommendations were given to these 264 patients. The majority of recommendations were off-label treatments (*n* = 248 of 367; 67.6%) and referral to accessible clinical trials (*n* = 67; 18.3%). In addition, recommendations comprised 52 in-label therapy-options (14.2%). The most commonly recommended treatment categories were single-agent targeted therapy (TT, *n* = 159 of 367; 43.3%), immune checkpoint blockade (ICB, *n* = 102; 25.1%), and combination therapies (CT; *n* = 92; 25.1%; Appendix A). To evaluate the clinical utility of molecular and genetic alterations, the MTB recommendations were categorized according to levels of molecular evidence (Appendix A). These were based on our initial definition [19] and were refined and harmonized for use in MTBs in Germany [53]. The majority of recommendations were based on molecular evidence from clinical trials or cohorts in the same (m1; *n* = 166 of 367; 45.2%) or in a different tumor entity (m2; *n* = 92; 25.1%) while a smaller fraction was based on molecular in vitro or in vivo evidence (m3; *n* = 101; 27.5%) or a biologic rationale (m4; *n* = 4; 1.1%; Appendix A).

Treatments were implemented in 76 of the 264 patients receiving a therapy recommendation (28.8%). The distribution of recommended therapies by type of treatment, implementation status and outcome is shown in Figure 2. Implemented off-label treatment recommendations were based on results of RMA (tNGS: *n* = 18 of 82; 22.0%; IHC: *n* = 17; 20.7%) and EGA (*n* = 12, 14.6%, Appendix A). Common causes for non-implementation were conditional recommendations designated to the future and not captured during the course of study (i.e., patients with stable disease under current therapy; *n* = 44 of 188; 23.4%), medical reasons (*n* = 43; 22.9%), and patient death (*n* = 35; 18.6%, Appendix A).

### 3.4. Clinical Outcome

Of the 76 patients that pursued an MTB treatment recommendation, 19 patients experienced stable disease (SD; 25.0%), 17 a partial response (PR; 22.4%), and five achieved a complete remission (CR; 6.6%), resulting in an overall objective response rate (ORR) of 4.5% (22 of 488 patients) and an overall disease control rate (DCR) of 8.4% (41 of 488 patients). Of the 41 patients with disease control, 30 received off label-treatments (30 of 488; 6.1%) and 23 received off-label treatments that were strictly based on a molecular rationale (23 of 488; 4.7%). To further validate the clinical significance of patients experiencing SD, we compared the PFS of the previous line of therapy (PFS1) to the PFS of the treatment recommended by the MTB (PFS2). A PFS2 to PFS1 ratio of 1.3 or higher is established as marker indicating benefit of a molecular-guided treatment in advanced cancer patients [17,54,55,56,57,58]. According to the modified progression-free survival ratio (mPFSr) [59], we excluded three patients from this analysis. One patient was removed after converting PFS1 times below two months into modified PFS1 intervals of two months to adjust for false positives, and two patients were excluded because PFS1 was not assessable. Of the remaining 20 patients, 17 fulfilled the criteria receiving strictly molecular-matched off-label treatment, and if SD, with mPFSr ≥ 1.3, resulting in an adjusted DCR of 3.5% (17 of 488 patients). Detailed clinical information of patients achieving at least stable disease and receiving off-label therapies between March 2017 and December 2018 (*n* = 16) are shown in Table 3. The corresponding data for 14 patients who had achieved at least stable disease between March 2015 and February 2017 were reported previously [19]. Two patients experiencing exceptional treatment response are presented in two case reports, medullary thyroid cancer and pleomorphic xanthoastrocytoma, in Appendix A and Case reports [60,61].

In order to assess the impact of the MTB on overall survival (OS) we used a Kaplan–Meier estimate to analyze survival of patients from first MTB presentation. To reduce possible confounders, we only included patients with stage IV malignancies (*n* = 340) and excluded patients who had died prior to therapy initiation (*n* = 53) or who showed disease control at the time of recommendation (*n* = 37). We observed a significant OS-benefit in 73 patients who pursued the recommended therapy (18 months; 95% CI, 11 to 30 months), compared to 100 patients who did not implement the treatment recommendation (OS = 8 months; 95% CI, 7 to 12 months; *p* = 0.008) or 167 patients who did not receive a recommendation (OS = 8 months; 95% CI, 7 to 12 months; *p* = 0.003).

## 4. Discussion

In this retrospective case series, we analyzed a total of 488 patients with mostly stage IV malignancies (*n* = 442; 90.6%), who were consecutively referred to a single institution MTB between March 2015 and December 2018, mostly because of progression to standard of care treatment (78.1%) or rare tumor entities (10.5%, Table 1). Based on entity-specific and entity-independent diagnostic SOP, we recommended individual molecular tests to 99.8% of the patients (*n* = 487) with a high implementation rate of 80.7% for diagnostic recommendations. The results enabled the MTB to issue individual treatment recommendations to more than half of the patients (*n* = 264; 54.1%) of which almost one third (*n* = 76 of 264; 28.8%) received the respective therapy. The overall ORR to recommended therapies was 4.5%, and the overall DCR was 8.4%. Other MTBs reported similar rates of treatment recommendations and implementation as well as DCRs: Baltimore [14] (24%, 16%, and 9%), Cleveland [15] (49%, 11% and 3.2%) and a recent report from Vienna [16] (54%, 23% and n.a., respectively). An ORR of 4.5% for a study in the oncology field is low. However, this number must be put into perspective given the challenging patient cohort. Most patients had metastatic cancer and were beyond standard of care treatment. Non-implementation of treatment was often related to patient death (18.6%) or bad general condition (7.5%). We therefore believe that the low percentage of patients experiencing disease control still represents a substantial proportion profiting from the MTB, given the challenging clinical situation of these patients.

We believe that our approach of SOP-driven and stepwise molecular diagnostics is time- and cost-effective as the median turnaround time from the first MTB to the first treatment recommendation was 42 days and the majority of recommendations did not require additional expensive and time-consuming EGA (85.4%). Treatment recommendations were based on EGA-results in 14.6% of patients. This included patients who received upfront WES or RNA-seq, i.e., patients with carcinoma of unknown primary, rare malignancies, or young patients. If the patient’s general condition afforded sufficient time to complete the analysis, EGA was also performed as a second diagnostic step when RMA did not identify any molecular target. Underlining the clinical effectiveness of treatment recommendations, patients with stage IV malignancies who received recommended treatments showed a significant benefit in OS (18 months; 95% CI, 11 to 30 months), which is encouraging compared to patients without recommendation (8 months; 95% CI, 7 to 12 months; *p* = 0.003) or lack of implementation (8 months; 95% CI, 7 to 12 months; *p* = 0.008). However, despite the attempt to reduce imbalances between subgroups the validity of this survival analysis is limited. Due to the low sample size, implementation of propensity score matching between cohorts was not possible, and thus potential confounders are beyond control.

Compared to our initial proof of concept report including 198 patients in 2018 [19] we were able to maintain constant rates of treatment recommendations (55.1% vs. 52.5%), while the treatment implementation rate remained relatively low (28.8% vs. 31.7%). Common causes for non-implementation were medical reasons (22.9%) and patient death (18.6%). We hypothesize that a streamlined workflow to grant access to off-label treatments using consented criteria would allow a higher proportion of patients to receive the recommended treatment in due time. Therefore, the MTB will in the future provide applications to health care providers. These will be sent with the corresponding treatment recommendation to the treating physician. Compared to our previous analysis, patient outcomes remained stable over time (DCR 7.6% vs. 9.6%), while case numbers per 90-min MTB meeting increased significantly (18.0 vs. 8.1). The efficiency of the MTB workflow was improved by assignment of patients to an entity-expert at least four days prior to first discussion, establishment in 2017 of a regular process to update diagnostic SOP, generation of automated reports for WES data within two days after receiving the raw sequencing data, communication of detailed diagnostic results to the entity-expert prior to MTB discussion, and the preparation of draft recommendations by the entity-expert to discuss and consent during the meeting (Appendix A).

Our report is in line with prospective precision oncology trials that demonstrated the impact of molecular driven therapies on patient outcome [11,12,13,63] underlining the importance of upscaling MTBs in order to facilitate treatment access to cancer patients who lack standard treatment options. A survey from van der Velden et al. revealed that in the Netherlands < 50% of hospitals and only 5% of non-academic hospitals had access to an MTB in 2017 [64]. To address this medical need, we opened the MTB to patients treated by external hospitals and private practice oncologists in June 2016. External referrals to the MTB increased from an average of 1.1 patients per session in 2016 to 1.7 in 2018. To further grant MTB-access to more cancer patients and to reduce discrepancies in care, related to diverse bioinformatics workflows [65] and heterogeneous standards for interpretation of molecular aberrations [66], the comprehensive cancer centers of southwest Germany upscaled referrals and harmonized their workflows and SOPs in 2020. This network initiative (Zentrum für Personalisierte Medizin, ZPM, Baden-Württemberg, Germany) also established a digital cloud that collects molecular diagnostic results and clinical follow-up data. This data will be used to identify both positive and importantly also negative correlations between molecular biomarker-driven therapies and outcomes. Accessible and effective drugs or clinical trials may therefore be identified easier and faster. Health care providers expressed an intrinsic interest to support and fund the establishment of the German ZPM-network since the increasing number of off-label requests from oncologists [67] will be streamlined through specialized MTBs allowing a harmonized and fast decision process with evidence based recommendations (Appendix A).

An important goal of the MTB is to improve access to available and accessible molecularly stratified trials for cancer patients. In close collaboration with the early clinical trial unit (ECTU) on site, the Freiburg MTB hosts a large number of innovative phase I and II clinical basket studies. Potential MTB-candidates are pre-screened at the earliest convenience for trials available on- and off-site to provide rapid access to these attractive molecular trial options. Compared to our initial report in 2018 we were able to increase the trial recommendation rate from 12.5% to 20.2%, a high rate comparable to recently published reports from Paris [68] (13.2%) and London [69] (20%). Based on informative MTB patients with exceptional responses, we were able to develop and initiate two investigator-initiated early clinical trials, ATLEP (DRKS00013336) and SORATRAM (DRKS00015849).

One of the major challenges of an MTB is to keep up with the dynamic field of predictive biomarkers for targeted and immuno-oncology treatments. For instance, the era of ICB opened up a new field of research to identify patients who would benefit from treatment outside the approved indications such as melanoma, lung, and renal cell carcinoma. In our cohort, established predictive biomarkers like expression of PD-L1 [70], MSI [71,72] and TMB [73,74,75] directed off-label ICB-mono or ICB-combination treatment recommendations in 108 cases (22.1%) of which 28 (25.9%) received therapy, resulting in a subgroup DCR of 57.1% (16 of 28 patients). Treatment response was associated with a positive IO-panel in 11 of 28 cases (39.3%) versus a negative IO-panel in five patients (17.9%; Appendix A). Interestingly, all seven patients with anaplastic thyroid cancer had a positive IO-panel with a TPS-Score ≥ 5, suggesting immune hot tumors. Indeed, all but one patient experienced disease control including two complete remissions. Of note, the positive results of the Keynote-158 study [71,73] led to a tumor-agnostic FDA-approval for pembrolizumab in June 2020 [5] reflecting one essential aspect of the MTB, namely, to provide patients early access to promising drugs not yet approved.

The identification of potential novel biomarker-driven drug treatments and therapy studies relies heavily on the expertise and knowledge of the MTB panel members. Hence, they face the challenge of critically examining the large, fast-moving literature of precision oncology to provide patients with the best possible molecular advice. Therefore, our interdisciplinary team is divided into entity-experts with strong expertise in molecular oncology for specific entities, molecular pathologists, medical bioinformaticians, and translational scientists.

A multitude of anti-cancer drug combination therapies have been introduced by the FDA since the initial approval of trastuzumab and pertuzumab in combination with the chemotherapeutic agent docetaxel in 2012 [76], increasing the probability and magnitude of treatment response as well as circumventing primary or acquired escape mechanisms. Accordingly, recommendations for combination therapies have increased from 18.3% to currently 28.5% since our first report in 2018. A total of 70 patients (14.3%) were recommended a combination therapy of which 21 (30%) received the respective treatment resulting in a subgroup DCR of 52.4% (11 of 21 patients, Figure 2). Of note, none of the implemented treatments were discontinued due to toxicity. Again, this indicates the success, but also the increasing complexity of modern anti-cancer therapy, necessitating structured molecular diagnostics and treatment recommendations.

## 5. Conclusions

This retrospective analysis of 488 patients included in the MTB Freiburg from 2015 to 2018 demonstrates a successful transition from proof-of-concept to routine clinical service. Despite significant upscaling processes, we confirm that molecular-guided precision oncology remains effective for a small but relevant proportion of advanced cancer patients that are beyond standard-of-care treatment.

## Figures and Tables

**Figure 1 cancers-13-01151-f001:**
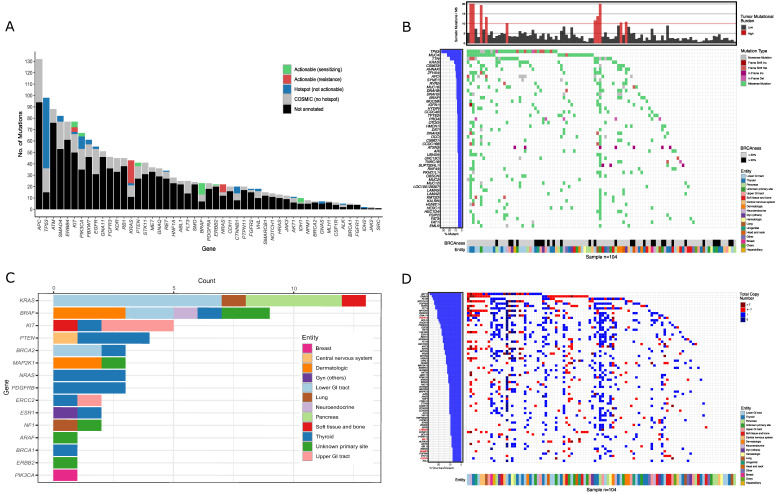
Results of sequencing: (**A**) The bar plot depicts the number of sequence variants detected in tumor DNA using the 48-gene panel for 221 patients. Colors indicate non-targetable COSMIC- (grey) and non-targetable hotspot mutations (blue). Actionable variants are shown in green (drug-sensitizing) and red (drug-resistance) based on the OncoKB classification. (**B**) The heatmap depicts the 50 most frequently mutated somatic genes of the 104 patients analyzed by whole-exome sequencing (WES). The colors indicate tumor entities, type of mutation, tumor mutational burden and BRCAness-score (= AC3−signature). Only mutations with a variant allele frequency greater than 10% and a minor allele frequency less than 0.1% were considered. (**C**) The bar diagram depicts all mutations that were annotated as targetable by the OncoKB classification. The colors indicate tumor entities. (**D**) The heatmap depicts copy number variations of the most frequently affected oncogenes. The colors indicate tumor entities and the total copy number per oncogene. Gene copy number gains that were annotated as targetable by the OncoKB algorithm are depicted in red.

**Figure 2 cancers-13-01151-f002:**
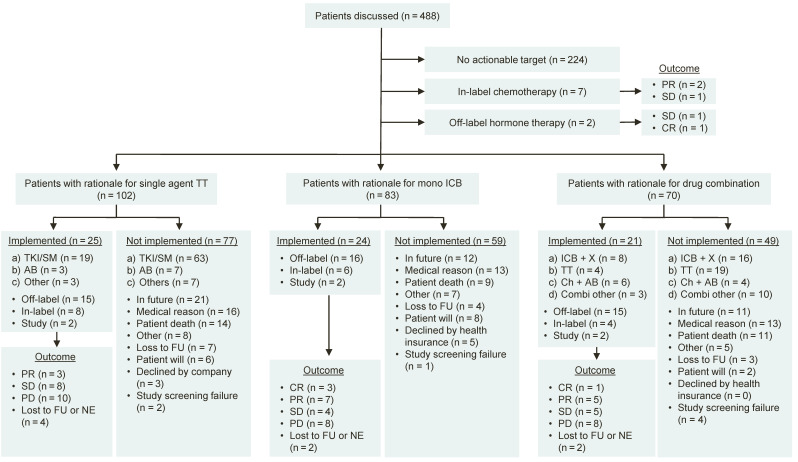
Flow diagram of patients discussed at the MTB. Responses were determined according to Response Evaluation Criteria in Solid Tumors (RECIST) version 1.1.; (a–d) represent various drug categories as indicated. PR = Partial remission. SD = Stable disease. PD = Progressive disease. ICB = Immune checkpoint blockade. TT = Targeted therapy. TKI = Tyrosine kinase inhibitor. SM = Small molecule. AB = Antibody. ICB + X = Combination of immune checkpoint blockade with another drug. Ch + AB = Combination of chemotherapy with an antibody. FU = Follow-up. NE = Not evaluable.

**Table 1 cancers-13-01151-t001:** Patient characteristics.

Characteristic	No.	(%)
Total	488	
Sex		
Female	230	(47.1)
Male	258	(52.9)
Median Age	54	range (1–88)
Patients with Solid Tumors: Stage at Presentation	470	(96.5)
Metastatic Disease	383	(81.5)
Localized Disease	86	(18.3)
Complete Remission	1	(0.2)
Tumor type		
Lower GI tract	68	(13.9)
Pancreas	50	(10.2)
Upper GI tract	42	(8.6)
Central nervous system	45	(9.2)
Unknown Primary Site	37	(7.6)
Hepatobiliary	30	(6.1)
Thyroid	30	(6.1)
Soft tissue and bone	37	(7.6)
Gyn (others)	18	(3.7)
Head and neck	19	(3.9)
Breast	21	(4.3)
Urogenital	12	(2.5)
Ovary	12	(2.5)
Dermatologic	18	(3.7)
Hematologic	17	(3.5)
Lung	16	(3.3)
Neuroendocrine	10	(2.0)
Other	6	(1.2)
Previous Lines of Therapy	2.05	(0–12)
0	66	(13.6)
1	152	(31.2)
2 to 3	192	(39.2)
>3	79	(16.2)
Unknown	1	(0.2)
Reason for Referral		
Progression to standard of care treatment	381	(78.1)
Rare Tumor	51	(10.5)
Young Age	29	(5.9)
Unknown Primary Site	20	(4.1)
Other	7	(1.4)

**Table 2 cancers-13-01151-t002:** Results.

Recommendations	No.	(%)
Meetings	95	
Case Discussions (per patient average)	499	(2.5)
Recommendations	1411	
Diagnostic	762	(54.0)
Treatment	367	(26.0)
No treatment recommendation	224	(15.9)
Conditional treatment recommendation	58	(4.1)
Patients with diagnostic recommendations	487	(99.8)
Routine molecular analysis	485	(99.4)
Extended genetic analysis	183	(37.5)
Rebiopsy	40	(5.2)
Other	14	(1.8)
Patients with Treatment recommendations	264	
Not implemented	188	(71.2)
Implemented	76	(28.8)
Stable disease (off-label)	19 (13)	(25.0)
Partial response (off-label)	17 (12)	(22.4)
Complete remission (off-label)	5 (5)	(6.6)
Disease control rate (off-label)	41 (30)	(8.4)

**Table 3 cancers-13-01151-t003:** Patients with disease control under off-label treatment since 2017.

Cancer Type	Rational for Treatment Recommendation	Board Recommendation	EL	L	R	PFS2 (Week)	PFS1 (Week)	PFSr	Outcome
Adrenocortical Carcinoma	Positive IO-Panel (TPS 2%, CPS 3, IC-Score 0). Chromosomal instability (CNV in 16 oncogenes and 28 tumor suppressor genes)	Study: Nivolumab for rare cancers (NCT02832167)	m2C	off	SD	36	37	1.0	SD for 36 weeks
	Tumor mutational burden high (11.07/Mb)	Pembrolizumab	m2C	off	SD	53	3	17.5	SD for 53 weeks
CRC	Positive IO-Panel (TPS 5%, CPS 10, IC-Score 1).	Combination of atezolizumab and cobimetinib	m2C	off	PR	76	11	6.9	PR for 76 weeks then switch to best supportive care
CUP	Signet ring cell carcinoma of unknown primary site with chromosomal Instability incl. a high copy number gain in EGFR (×338).	Combination of cetuximab and FOLFIRI	m2A	off	SD	25	31	0.8	SD for 25 weeks the PD with new ascites. Switch to Combination of paclitaxel and ramucirumab
Histiocytosis	Erdheim Chester disease with BRAF-V600E mutation.	BRAF-inhibition with vemurafenib or dabrafenib	m1B	off	PR	>133	51	>2.6	Initial treatment with vemurafenib. Due to toxicity switch to dabrafenib after 10 weeks. Very good PR in cMRI after 39 weeks therefore switch to maintenance therapy with peg-INF. PR still ongoing
Meningioma	Anaplastic meningioma. IHC shows strong staining for somatostatin on 100% of tumor cells	Octreotide	m3	off	SD	18	n.a.	n.a.	SD for 18 weeks then PD with new resection. PFS1 n.a. since no prior systemic therapy received. Only surgery and radiation therapy
Mesothelioma	Despite negative IO-Panel (TPS < 1%, CPS 10, IC-Score 1) data show response to checkpoint-Inhibition [62]	Pembrolizumab	m1C	off	SD	32	56	0.6	SD for 32 weeks then patient death
PXA	Pleomorphic xanthoastrocytoma with BRAF V600E mutation	BRAF-Inhibition with combination of dabrafenib and trametinib	m1C	off	CR	>64	n.a.	n.a.	CR for 64 weeks and ongoing. PFS1 n.a. since no prior systemic therapy received. Only surgery and radiation therapy.
Prostate	Deleterious BRCA2 mutation.	Platinum-based chemotherapy followed by olaparib	m2A	off	SD	31	7	4.5	SD for 31 weeks then PD
Salivary	IHC shows strong androgen receptor staining in 90% of tumor cells.	Combination of degarelix and bicalutamide	m1C	off	CR	66	n.a.	n.a.	CR for 66 weeks then PD with new metastasis. PFS1 n.a. after resection (R0) and adjuvant radiochemotherapy
Sarcoma	WES shows BRCAness of 29%	Combination of olaparib and trabectedine	m3	off	SD	6	4	1.3	SD in first staging after 2 weeks. Due to pain initiation of palliative radiation therapy with consecutive esophagitis and pancytopenia resulting in death at 6 weeks
Thyroid (anaplastic)	Positive IO-Panel (TPS 5%, CPS 9, IC-Score 1). Study availability	Combination of pembrolizumab and lenvatinib	m2C	off	PR	>53	16	>3.3	PR for 53 weeks and ongoing
(anaplastic)	Positive IO-Panel (TPS > 80%, intratumoral TILs)	Pembrolizumab	m2C	off	CR	25	7	3.0	CR but died due to pulmonary bleeding at 21 weeks
(anaplastic)	Positive IO-Panel (TPS 5%, intratumoral TILs). Chromosomal instability with strong amplification of *PDGFRA* (×28) and *PDGFB* (×29)	Combination of pembrolizumab and lenvatinib	m2C	off	CR	83	16	3.9	CR for 83 weeks and ongoing. Lenvatinib was discontinued after 52 weeks.
(anaplastic)	RNA-Seq shows upregulation of FGFR3 signaling pathway	FGFR3-Inhibition with lenvatinib	m3	off	SD	11	n.a.	n.a.	Good clinical response with regressive local relapse and significant clinical improvement. Therapy was discontinued after 14 weeks due to weight loss, then PD and death.
(medullary)	RET Met918Thr Mutation	Selpercatinib	m1A	off	PR	35	15	2.3	Radiologic PR. Serologic decrease of calcitonin from 8554 pg/mL to 12 pg/mL. Response ongoing.

Forty-one out of 488 patients showed disease control under the recommended treatment. Progression free survival (PFS1, PFS2) was determined according to Response Evaluation Criteria in Solid Tumors (RECIST) version 1.1. Median PFS2 (53 weeks; range 6–238 weeks) was significantly longer than median PFS1 (16 weeks; range 3–16 weeks; *p* = 0.003). Here we list the yet unpublished off-label responders (16 patients) that were discussed between March 2017 and December 2018. Relevant diagnostic results leading to a board recommendation, level of evidence (EL), label (L), treatment response (R), progression free survival (PFS1, PFS2), progression free survival ratio (PFS2/PFS1 = PFSr) and outcome are shown. PFS1 and PFSr could not be evaluated for four patients since they did not receive a prior systemic treatment. PR: partial response. SD: stable disease. n.a.: not applicable. CUP: cancer of unknown primary. IO-Panel: Immuno-oncology panel. TPS: Tumor Proportion Score. CPS: Combined positive score. IC-Score: Immune cell score. TILs: Tumor infiltrating lymphocytes. CNV: Copy number variation. PXA: Pleomorphic xanthoastrocytoma.

## Data Availability

Source data are provided with this paper. All remaining relevant data are available in the article, Appendix A, or from the corresponding author upon reasonable request.

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
