# Peer review of "Transitioning the Molecular Tumor Board from Proof of Concept to Clinical Routine: A German Single-Center Analysis"

_cancers, 2021, doi:10.3390/cancers13051151_

Round 1

Reviewer 1 Report

This is a thoroughly documented study with clinical relevance. The information provided is helpful as an orientation for molecular tumor boards in other institutions and can guide the implementation of such boards.

I want to make a few comments on potential minor revisions.

a) could you briefly describe the evaluation/scoring of CD3, CD4, CD8 and PD-1 in your IO-panel?

b) in Table 3 intratumoral TILs are mentioned as a rational for therapy recommendations in two anaplastic thyroid carcinomas. How did you quantitate TILs? Did you establish a cut-off for TILs for immunotherapy?

c) Whole exome sequencing and RNAseq: performed with FFPE tissue or unfixed, frozen/fresh tissue?

d) did you use a lower cut-off of tumor cell content of tissues for selection of suitable samples for sequencing?

e) it would be helpul to include the list of genes of the targeted NGS panels in the Supplements, at least for the presumably custom 8 and 15 gene panels.

f) could you specify "technical reasons" in line 217 (reasons for lack of implementation of diagnostic recommendations)

g) Sorry, I do not understand in Table 2 the part "Patients with diagnostic recommendations" with Routine Pathology in 458 cases. Routine pathology is not the recommendation of a molecular tumor board. It is performed prior to submitting a case to the board. 

h) in Table 3 the word "sandostatin" is most likely a mistake. I suppose it should be "somatostatin"?

Reviewer 2 Report

Congratulations for the prospective registry of the MTB data.

Overall, I think that the wording is too optimistic, with many far-reaching statements that must be tempered. No doubt that consensus guidance for molecular testing, results interpretation and drug matching have a huge impact in patient care. But the clinical utility as direct patient benefit attributed to non-standard diagnostics (EGA) and access to emerging drugs is still very small.

In many instances I do not understand the numbers presented by the authors. For example, 76 recommendations for molecularly-guided therapies were implemented (25 + 24 +21?), representing 28.8% (what is the denominator)? There are so few actionable gene alterations in this (mostly GI/rare tumors) cohort, so what is the value of expanded molecular tetsing? Interesting to look at the data beyond genomics as well - how many TRK positive by IHC were found to be fusion positive, for example? Or how many IO-positive cases by tumor type - and linked response to ICB?

The survival analysis is extremely biased by patient selection and immortal time. If you believe that OS gain is really attributed to the molecular-guided therapy (73 or 76 patients?), I suggest propensity score matching to tumor type, treatment line, performance status. I recommend you presnt the determinants of a match that was pursued vs. not implemented. This is far more insightful than a biased KM. 

In terms of methods, I would like to understand better how was TMB calculated, criteira for 10 mut/MB cut-off (normalization with FoundationOne?), how was BRCAness estimated, etc.

In Figure 1, why you separate actionable vs. Hotspot? Not annotated means real variant (as per bioinformatics filters) and predicted functional + VUS? What was reported to clinicians?

Reviewer 3 Report

The authors present the results of their MTB along with clinical efficacy of matched therapy. There are 3 important points that need to be corrected:

1) The conclusions of the abstract and the manuscript are not supported by the data. The authors Indeed claim that: "Our data demonstrate the feasibility and effectiveness of molecularly guided personalized therapy for cancer patients in a clinical routine setting showing a robust and durable disease control rate over time." First the results show a DCR of only 8.4% when taking the overall patient population (which is the right thing to do). In addition, disease stabilization is not a meaningful Endpoint since it dépends of the natural history of the cancer that obviously dépends on tumor types. There fore, only ORR is meaningful. The authors found an ORR of 4.5% which is very low although in line with previous report.

2) When calculating their DCR, the authors have included patients for which we cannot tell whether matched therapy was the cause of the response, especially for patients who received a drug combination (e.g. CUP patient who received FOLFIRI + cetuximab). Theya lso included patients who actually did not receive matched therapy (e.g. patient with PD-L1- mesothelioma for whom the decision to treat with anti-PD1 did not depend on the result of the MTB). All these patients should be excluded from the DCR and ORR determination.

3) The OS curves and the comparison are misleadinging. The comparison of OS in patients with rec. pursued (18 months) versus patients with rec. not pursued (8 months) should not be made, since there might biaises related to the Reason why patients did not receive matched therapy. The OS analysis should be removed from the paper, since conlcusions are not robust.

Round 2

Reviewer 3 Report

I still do not feel confortable with this paper. My commets were not fully taken into consideration, especially regarding the denominator that should be taken into consideration for calculating the ORR, the lack of relevance of DCR, the fact that patients who exeprienced a repsonse but who did not clearly received matched therapy were not excluded from the numerator. The authors Added here the PFS ratio data but without mentioning médians for PFS1 ad PFS2, nor how PFS1 and PFS2 were calculated (according to RECIST both?). Overall, this paper shows how limited is the utility of Molecular profiling, except for very few patients.

Round 3

Reviewer 3 Report

The authors still did not taken into consideration one point I mentioned in my 2 previous reviews concerning DCR. Indeed, DCR is not an appropriate end point , since disease stabilization might only be the consequence of early disease evaluation. As mentioned, I think only ORR data should be presented.
